# Development and Psychometric Properties of MisoQuest—A New Self-Report Questionnaire for Misophonia

**DOI:** 10.3390/ijerph17051797

**Published:** 2020-03-10

**Authors:** Marta Siepsiak, Andrzej Śliwerski, Wojciech Łukasz Dragan

**Affiliations:** 1Faculty of Psychology, University of Warsaw, 00-183 Warsaw, Poland; wdragan@psych.uw.edu.pl; 2Institute of Psychology, University of Łódź, 90-136 Łódź, Poland; andrzej.sliwerski@uni.lodz.pl

**Keywords:** misophonia, decreased sound tolerance, misophonia questionnaire, MisoQuest, sound sensitivity

## Abstract

*Background*: Misophonia is a condition related to experiencing psychophysiological sensations when exposed to specific sound triggers. In spite of progress in research on the subject, a fully validated questionnaire assessing misophonia has not been published yet. The goal of this study was to create and validate a new questionnaire to measure misophonia. *Methods*: MisoQuest is based on the diagnostic criteria proposed by Schröder et al. in 2013, with minor changes implemented by the authors of MisoQuest. A total of 705 participants took part in the study, completing the online questionnaires. Exploratory Factor Analysis (EFA) and analyses using the Item Response Theory (IRT) were performed. Internal consistency was evaluated with Cronbach’s alpha. *Results*: The reliability of the MisoQuest was excellent (α = 0.955). The stability at five weeks was strong. There was a significant difference in results between people classified as those with misophonia and those without misophonia. *Conclusions*: MisoQuest has good psychometric values and can be helpful in the identification of misophonia. A deeper analysis showed that certain triggers might be more specific for people with misophonia. Consideration of violent behavior in response to misophonic triggers as a symptom of misophonia was undermined.

## 1. Introduction

Misophonia, or Selective Sound Sensitivity Syndrome, is a type of decreased sound tolerance characterized by increased, autonomic nervous system response to specific triggers [1,2,3]. The phenomenon of this specific sound sensitivity was first described by Jastreboff in 2001 [4]. However, since then the understanding of misophonia has evolved.

People with misophonia while being exposed to certain auditory triggers experience strong, immediate, unwanted emotions and physiological reactions such as increased heart rate, sweating, and muscle tension. The dominant, but not exclusive, emotions related to misophonia are anger, disgust, extreme irritation, and anxiety [3,5,6]. In contrast to the less-discussed phonophobia [7], misophonia is not associated with experiencing fear of the sound while it is occurring [8,9], excluding it from classification as a phobia. However, the definitions and classification of these phenomena are still being discussed; for example, Jastreboff [2] classified phonophobia as a type of misophonia. Among the most aversive triggers are sounds made by human mouths or noses, such as chewing, breathing sounds, lip-smacking, crunching, or sniffing. Some researchers [6] noted that the sound of a baby crying orhigh-pitched voices can also trigger misophonic reactions; however, the study by Kumar et al. [10] showed that these kinds of sounds evoke significantly different reactions on psychophysiological and neuropsychological levels than do sounds related to eating. They found that in people with misophonia, sounds of eating, breathing, or drinking (trigger sounds) are related to abnormal activation of the anterior insular cortex (AIC) and abnormal functional connectivity of this structure with brain areas responsible for emotion processing and regulation. Other sounds produced by people but not directly by the human body, for example, pen clicking, rustling, tapping, typing, shuffling, and environmental sounds, can also be misophonic triggers [5,8]. In misophonia, according to the current state of knowledge, the acoustic characteristics of sounds have no impact on emotional arousal [3] or are secondary [1,2]. The type of aversive triggers differ from person to person and the reaction towards trigger sounds can depend on many factors, such as assessment of the sound, personal experience, social context or psychological profile [2].

Misophonia is not yet classified by any official diagnostic system, either the Diagnostic and Statistical Manual of Mental Disorders, Fifth Edition (DSM-V) or the International Classification of Diseases, Eleventh Edition(ICD-11), but it seems that despite previous beliefs, misophonia is itself a separate disorder, rather than a symptom of other disorders [11,12]. Schröder et al. published the first complete proposition of diagnostic criteria for misophonia in 2013 [8]. According to these criteria, misophonia is diagnosed when:An individual experiences a strong, immediate somatic reaction that begins with irritation or disgust which quickly turns into anger in the presence of or in the expectation of aversive sounds. The anger makes a person feel out of control, sometimes leading to aggressive behavior;The person assesses these reactions as disproportionate to the situation;Due to the consequences of unpleasant experiences caused by certain sounds, the person, if possible, avoids situations in which the trigger is expected or struggles with high discomfort in its presence. Therefore, the condition has a significant, negative impact on the person’s life;The avoidance and emotional reaction to certain triggers cannot be better explained by other disorders, such as post-traumatic stress disorder or obsessive-compulsive disorder.

Despite significant progress in the field of misophonia, this description still seems to be valid. Currently, however, most scientists agree that all kinds of sounds may provoke a misophonic reaction, not only those produced by humans [3]. New data has led to the creation of other criteria for diagnosing misophonia, broadened in terms of the potential sensory modalities of the triggers as well as the type of emotions and body sensations that occur. Some researchers claim that the aversive reaction can be triggered not only by sounds, but also by all kinds of triggers, such as smell, movement, or touch. The immediate physical reflex response, such as muscle constriction, was suggested to be the initial reaction in misophonia [13]. Nevertheless, such cases are less documented.

Some questionnaires for measuring misophonia have been developed and used in several studies. The Misophonia Questionnaire (MQ), developed by Wu et al. in 2014 [14], is the most detailed and widely used [12,15,16,17]. The data was gathered on a group of undergraduate students of psychology and showed good psychometric values. The MQ has good internal consistency. However, it is not clear whether the questionnaire measures misophonia or, rather, some other, more general phenomenon because its external validity was verified with a questionnaire which measures general sound sensitivity. The MQ was developed before the core details of misophonia were agreed upon, and therefore may benefit from refinement.

Another widely used questionnaire, the Amsterdam Misophonia Scale (A-MISO-S), was developed by Schröder et al. [8].This scale was one of the first attempts to measure misophonia and, like the MQ, played an essential role in exploring this phenomenon. Its psychometric values have not yet been presented though. A-MISO-S was adapted from the Yale–Brown Obsessive Compulsive Scale at a time when misophonia was still strongly associated with OCD. Subsequent research did not confirm this association and made the specific characteristics of misophonia clearer [11,12]. However, the findings from other research on the relationship between misophonia and OCD are not consistent, showing certain common traits to these two conditions [16]. Other scales which measure misophonia include the Misophonia Assessment Questionnaire (MAQ), created by Johnson and revised by Dozier and published on misophoniatreatment.org [18], The Misophonia Physical Sensation Scale, created by Baumanand published on misophoniatreatment.org [19], or the Misophonia Activation Scale (MAS-1), created by Fitzmauriceand published on misophonia.co.uk [20]. All these scales have been used in research on misophonia, but none have been validated or published in peer-reviewed journals.

The growing interest of researchers in the field of misophonia has increased the number of studies on the subject. Despite significant improvements in knowledge about misophonia, the problem of generalization and comparison between research results emerges, which is exemplified by the recent discussion between two prominent misophonia researchers—Schröder and Kumar [21,22]. The criteria used to diagnose misophonia in research often differ. Therefore, results from studies cannot be easily compared with each other. Consequently, there is a need to develop a validated tool for diagnosing misophonia.

At the moment, there is no fully validated questionnaire available that measures misophonia based on the current understanding of the disorder. This study aimed to develop and validate a new questionnaire to assess misophonia. The construction of the tool was based on the criteria specified by Schröder et al. [8]. Thus, specific misophonia-like sensitivities across other modalities, such as misokinesia or sensitivity to certain fabrics, were not included. As well as standard statistical methods, the Item Response Theory (IRT) was used to assess the diagnostic value of each item and, thus, to determine which criteria best differentiate people with misophonia from those without this condition. Some statistics were calculated separately for participants with education in music because it was hypothesized that they can perceive sounds and understand questions related to sounds differently [23,24,25] and, therefore, their MisoQuest results would differ. Similarly, the group of participants who self-reported a psychiatric diagnosis was distinguished in the study due to potential relationships with misophonia.

## 2. Materials and Methods

### 2.1. Ethical Approval

The study is a part of a research project named “Psychological and psychophysiological correlates of misophonia (decreased sound tolerance)”, conducted at the Faculty of Psychology at the University of Warsaw, Poland. The Ethics Committee at the faculty approved the project (document number: 12/5/2018). The study was conducted on-line. All participants gave informed written consent before taking part in the study and also accepted the General Data Protection Regulation (GDPR) policy.

### 2.2. Questionnaire Construction

The questionnaire was developed in two phases (on two different samples). The data collection period spanned between May and December 2018. Firstly, the pool of items was created following the criteria of Schröder et al. [8]. Items were arranged in seven domains extracted from the criteria (see Table 1): (1) reaction to specific sounds; (2) an occurrence of reaction; (3) emotional reactions; (4) control of emotional reactions; (5) attitude to own reactions; (6) avoidance; (7) and daily functioning. For each domain, a set of items (questions suited to the domains) was created (ranging from 6 to 14 items per domain). Two psychologists with experience working with patients with misophonia assessed items in terms of comprehensibility and compliance with the criteria of misophonia. The first phase was initiated after incorporating their linguistic and theoretical corrections to the questions. The initial version of MisoQuest had 60 items. Because there were similar questions within each domain, verification of their power began by analyzing their consistency with the scale for the whole group as well as groups of individuals with and without misophonia-like issues (self-report). At this stage of the questionnaire development, we paid attention not only to the power of the questions but also to their content. We tried not to lose questions that can diagnose the entire spectrum of over-responsiveness to various sounds. The results were analyzed, and 21 items were used for the second phase. As a result of the second phase, 14 items were chosen for the final version of MisoQuest. To assess the occurrence of misophonia, the Schröder et al. [8] criteria were used. All participants were asked in the last part of the survey if they experienced symptoms characteristic of misophonia, and only those that met all the diagnostic criteria were deemed to have misophonia. During this phase, the participants were also asked whether they had been diagnosed by a specialist (i.e., not self-diagnosed) with any disorder.

### 2.3. Participants

During both phases of the development and validation of MisoQuest, 705 participants with and without sound issues participated in the study (383 and 322 participants respectively for each phase). The questionnaire was distributed online in support groups for people with misophoniaand various groups unrelated to either misophonia or other health issues, due to the limited number of people with misophonia. Participants were aged 18 to 68 (*M* = 32.09, SD = 8.58). In the second phase, seven participants under 18 years old were excluded from the analysis. Women constituted the majority of the respondents (86.2% and 80% respectively for each phase). Despite such large disproportions, no significant differences were found in Phase 1 between men and women in the presence of misophonia (*χ*^2^(2) = 3.53; *p* > 0.05). However, in Phase 2, the differences were significant. More women (*n* = 55; 21.8%) than men (*n* = 6; 9.5%) had misophonia according to their responses to the questions given at the end of the survey (*χ*^2^(1) = 4.88; *p* < 0.05). During the second phase of the study, the diagnosis of neurological and psychiatric disorders was validated by self-report (see Table 2). A group of musicians who practice from one hour to seven hours a day was also distinguished.

### 2.4. Data Management and Analytic Strategy

The construct validity of MisoQuest was examined via Exploratory Factor Analysis (EFA) using Principal Component Analysis (PCA). Internal consistency was evaluated with Cronbach’s alpha. Confirmatory Factor Analysis (CFA) is usually performed to validate potential questionnaire models obtained by EFA [26].

Because the final form of MisoQuest had only one factor and very high inter-item correlations and inter-total correlations, CFA was abandoned. Analyses using the IRT were performed instead. IRT models help to explain the relationship between latent traits (symbolized by *Ɵ*) and their manifestation in questionnaire responses. In simple terms, they help determine the precision of psychological scales. The stronger the trait is, the higher the probability of a correct response (answer in accordance with the key) [27]. However, IRT models assume that a scale’s items are not equally informative across the latent trait range. The informative value depends on two parameters of the item: difficulty and discrimination. Due to the Likert-type scale used in MisoQuest, we used the Graded Response Model (GRM) for polytomous data [28]. In this model, difficulty parameters indicate the threshold of a latent trait between answer choices and the discrimination parameter reflects the degree to which an item discriminates individuals across the latent trait range [29]. To compare the fit of the two models (constrained and unconstrained), the Akaike’s Information Criterion (AIC) [30] and Bayesian Information Criterion (BIC) [31] were calculated. To examine the concurrent validity of MisoQuest, bivariate analysis was conducted using the Student’s *t*-test for independent samples. Finally, a subset of the validation study (*n* = 97) was used to evaluate test-retest reliability at a five-week interval.

Data analysis was carried out with SPSS v25 (SPSS Inc., Chicago, IL, USA) and RStudio, with alpha criterionof 0.05. RStudio [32] was used to conduct IRT with the ltm, msm, and polycor packages [28] and also to conduct ROC/AUC analysis with the pRoc package [33]. All graphs in this paper were prepared using RStudio.

## 3. Results

An Exploratory Factor Analysis (EFA) was conducted on 60 items of the first version of the questionnaire, producing a Kaiser-Meyer-Olkin (KMO) index score of 0.971 and a statistically significant Bartlett’s test (*p* < 0.001). Following Kaiser’s criteria, the EFA provided a model composed of eight factors, accounting for 69.84% of the total variance explained. However, Cattell’s scree test indicated that only one factor lay above the debris (see Figure 1A), accounting for 50.05% of the total variance. Thirty-eight items had high loadings (>0.7), and ten items had low overall loadings (<0.5). At least two of the best items were selected from each of the seven domains (see Table 1). Two items were added to the first version of the questionnaire because they matched the criteria distinguished by other researchers [34]. The first of them concerned sounds only made by objects (Q13) and the second concerned violent reactions toward others caused by the trigger (Q38). As a consequence, a preliminary version of the questionnaire consisting of 21 items was created.

### 3.1. Psychometric Validation: Construct Validity

The second phase of the development of MisoQuest was prepared using an entirely different sample (*N* = 315). An exploratory factor analysis (EFA) revealed a KMO Index of 0.961. Bartlett’s test of sphericity was statistically significant (*χ*^2^ = 4952.663, *p* < 0.001). This analysis provided a factorial structure with two factors that accounted for 61.63% of the variance. However, the Cattell’s scree test indicated that only one factor should be taken into account (it explained 56.29% of variance; see Figure 1B). Twenty items were assigned to the first factor and only one item to the second factor (Q38 “As a child, I beat someone who made unpleasant sounds”). Standardized item-scale correlations and factor loadings are presented in Table 1.

### 3.2. Item Response Theory Analysis

The Graded Response Model [35] is the simplest model created to examine the psychometric properties of tools with multiple response formats (like Likert-type scales). Polytomous models are extensively used in applied psychological measurement to help reduce test length without losing valuable information. In the first step, we evaluated the fit of the unconstrained and constrained models (assuming not equal and equal discrimination parameters across items, respectively). The likelihood ratio test for the unconstrained model (AIC = 15,726.13; BIC = 16,122.79) was significantly better than for the constrained model (AIC = 15,904.54; BIC = 16,225.64). Models with lower AIC values are desirable, as they indicate a closer fit to a true model [36].

In IRT, the discrimination level, α, is considered high if its value is higher than one. Only one item (Q38), which was recognized as a separate factor in EFA, did not reach the low discrimination level. Most items had an α level higher than two, which means that these questions are very good in differentiating people with and without misophonia. Trace curves for each item (Figure 2) were visually inspected to determine whether each answer choice had an area of a latent trait that was most probable.

Trace curve analysis showed that for almost all items, the individuals responded very dichotomously. They unambiguously chose option one or five more often than the other options. This is probably due to the specificity of misophonia symptoms. For items Q4, Q12, Q23, Q26, Q29, Q32, Q35, Q40, Q47, Q49, and Q56, the neutral answer (choice three) was chosen less often than other item choices. This means that individuals either had moderate or strong symptoms or had none at all.

Based on the results of the EFA and IRT, 14 items were selected for the final version of MisoQuest. Figure 3 contains the item information curves divided into those that were selected for the questionnaire (Figure 3A) and those that were rejected (Figure 3B). The higher the curve in the chart, the greater the discrimination factor of the item. The more it is shifted to the right, the better it differentiates people with misophonia. Item Q4 has the lowest discrimination rate of the MisoQuest items, but it is a question that fewer people respond to using the key. It is also the only question that directly asks about sounds made by the human body. Furthermore, it is the best question according to EFA in terms of reaction to the specific sound criteria of misophonia. On the other hand, Q15 was rejected despite a very good discrimination level (*α* = 2.010), since many subjects responded to it according to the key. The Test Information Curve for the final version of the MisoQuest was estimated and indicates that the curve covers the desired range from −2 to +2, which is necessary for a clinical diagnostic tool. The accuracy of the measurement for subjects with misophonia for the entire scale in the range from −2*Ɵ* to 2*Ɵ* was 75.80 (94.35% complete information).

### 3.3. Reliability Analysis

Internal consistency of the MisoQuest was assessed by applying Cronbach’s alpha in both samples (utilizing only the final 14 items). The reliability of MisoQuest was excellent in both samples (α = 0.961, α = 0.955, respectively). It was found that most items correlated with the total score, with values ranging from 0.69 to 0.83 (see Table 1). This indicates the very high homogeneity of the questionnaire. For test-retest reliability, all participants from the second phase were asked to complete the test again. The analysis was carried out on 97 complete response pairs. The stability of the instrument at five weeks was strong. The interclass correlation coefficient (ICC) in a two-way random model in absolute agreement for the total scale was 0.84, 95% CI (0.78, 0.89). Even the test-retest reliability for every item was very high, ranging from 0.59 to 0.78 (80% of items had an ICC higher than 0.67).

### 3.4. Criterion-Related Validity

An independent *t*-test was utilized to evaluate the concurrent validity of MisoQuest. The means (SD) for MisoQuestare based on different diagnoses and are for musicians and non-musicians; details can be found in Table 3. There was a significant difference (*t*(306,855) = 21.65; *p* < 0.001; Cohen′s d = 2.13) in results between people classified as having misophonia (*M* = 65.72; SD = 4.3) and those without misophonia (*M* = 41.77; SD = 15.3). After exclusion of people with developmental, affective, and anxiety disorders, the difference between people with misophonia (*n* = 47; *M* = 66.04; SD = 3.66) and without misophonia (*n* = 206; *M* = 39.99; SD = 15.14) remained similarly significant, but slightly higher and stronger (*t*(250,296) = 22.04; *p* < 0.001; Cohen’s d = 2.37). Further analysis showed that people with any diagnosis (*M* = 52.87; SD = 13.84) obtained significantly higher results on the scale (*t*(111,322) = 3.9; *p* < 0.001; Cohen′s d = 0.52), than people with no diagnosis (*M* = 44.83; SD = 17.09). All diagnoses, such as autism, OCD, depression, and anxiety disorder, were merged into one factor because of the limited number of participants with a particular diagnosis.

Surprisingly, musicians (*M* = 43.55; SD = 13.89) got significantly lower average results (*t*(225,670) = 2.51; *p* < 0.05; Cohen′s d = 0.26) than did non-musicians (*M* = 47.64; SD = 17.78).

## 4. Discussion

The goal of this study was to develop and assess the psychometric value of MisoQuest, a new questionnaire for measuring misophonia or Selective Sound Sensitivity Syndrome. MisoQuest consists of 14 items organized within one factor. Its internal consistency is excellent and its stability over a five-week period is substantial. Criterion-related validity was preliminarily confirmed with a set of self-reported questions derived from the diagnostic criteria for misophonia [8].

There are a few characteristics that distinguish MisoQuest from other questionnaires that also measure misophonia. In contrast to the most common tool, A-MISO-S, and the third part of MQ, MisoQuest was not based on a questionnaire for assessing OCD, so it does not include items specifically associated with OCD such as time spent resisting thoughts about misophonic sounds. Rather, its items deal specifically with elements related to the current understanding of misophonia. There are also no questions about the time consumed by trigger sounds, as the answer can depend more on the respondent′s adaptive abilities, avoidance, way of life (e.g., working from home) or time of the year (e.g., Christmas time). In MisoQuest, the respondent is not directly asked about “misophonic” sounds, because it is not intended to assess the intensity of misophonia, but rather to screen for misophonia.

MisoQuest was constructed based on the misophonia criteria identified by Schröder et al. [8], with some modifications. Possible triggers were extended from sounds only produced by humans to all kinds of sounds and the item on aggressive behaviors was excluded. The role of anger is emphasized (items Q12 and Q19) due to the psychometric indications as well as the aim to capture extreme emotions, however, there is also a question on irritation (Q21) and a number of items deal with other unpleasant emotions (Q5, Q8, Q9, and Q16). In order to identify the type of trigger sounds, separate items related to sounds (only human vs. other sounds) were created.

According to the criteria, the emotional reaction of anger needs to be immediate (item Q19), so MisoQuest will not identify people who are irritated or feel uncomfortable only when the sound does not stop for a long time. This likely means that fewer people will be diagnosed with misophonia when assessed with MisoQuest than with MQ, but those diagnosed will be more homogeneous. It is not possible to draw conclusions about the prevalence of misophonia (measured by MisoQuest) from this study, as this was beyond its scope. Because of these characteristics, MisoQuest can be considered to be a good tool to measure misophonia as defined as a condition in which a person is triggered immediately by certain sounds and anger as a core, but not necessarily exclusive, emotion. Therefore, it is not claimed that MisoQuest should always be the first questionnaire of choice. The use of the tool should depend on the way misophonia is defined.

Our analysis also revealed some meaningful hints regarding the characteristics of misophonia which are mostly consistent with or complementary to the available data on this phenomenon.

One of the most interesting secondary findings of this study is the significance of the source of the misophonic triggers. In the first version of Schröder’s criteria for diagnosing misophonia, the only potential source of trigger sounds was human. In several studies, such sounds have been found to be the most aversive for people with misophonia. Nonetheless, a number of studies have shown that all kinds of sounds, not only human-related, can potentially provoke reactions in people with misophonia [2,13]. Therefore, in MisoQuest we included all kinds of sounds as possible triggers. The item regarding sounds made by humans had the most significant psychometric values in EFA. IRT analysis revealed that this question was the most specific for people with misophonia.

Interestingly, an item regarding reactions towards quiet, soft sounds had similar psychometric properties to the one about human sounds, which are usually soft as well. Thus, this is another indication that the source of trigger sound in misophonia might be more specific than is currently believed and more related to humans, as was first postulated in the diagnostic criteria for misophonia [8]. Nonetheless, this needs further exploration with more objective testing.

Another important discovery was the invalidity of a questionnaire item related to physical violence (Q38) and verbal violence (Q37 in the 60-item version). The inclusion of these items in a preliminary version of MisoQuest may be seen as controversial, but it was intentional. The occurrence of an act of violence in response to triggers among people with misophonia is mentioned somewhat anecdotally in certain publications, but can also be found in some non-validated scales for misophonia (e.g., the Misophonia Activation Scale). At the same time, the data shows that most cases of misophonia develop in childhood [5,8,9,11,37]. Here, participants were asked if they had assaulted anybody in their childhood because of unpleasant sounds made by this person. Questions were asked about the past, as asking about the present might be aggravating. Moreover, aggressive behavior in misophonia is sometimes reported as being present in children [38] (however, there is not enough data to draw definitive conclusions about its presence in adults).This study showed that there is no relation between recalled misophonia experiences in childhood and acts of violence—both IRT and EFA analysis indicated the inadequacy of this item. It is worth noting that some participants commented that this question surprised them and in their opinion, it has nothing to do with sound sensitivities or misophonia (both in the comments section of the survey and on the internet forum where the questionnaire was made available). However, because the term “childhood” was not defined for the participants of the study and data on the onset of misophonia among participants was not gathered, it is not possible to draw strong conclusions from these results regarding the relation between misophonia in childhood and acts of violence.

An interesting characteristic was found regarding the intensity of symptoms on the scale. People with psychiatric disorders (depression, obsessive-compulsive disorders, and anxiety disorders) together with autism, merged in one variable due to the limited number of participants with a second diagnosis, scored significantly higher on the scale than people without any diagnosis. It is worth pointing out that the relation between sound-sensitivity issues experienced by people with autism spectrum disorder and misophonia is not known. More data on the characteristics of sound sensitivities in people with psychiatric disorders and autism is needed. In this study, the difference between mean scores of people with psychiatric disorders and people with autism was only two points. However, after the exclusion of participants with misophonia (see the raw data are available in the Appendix A), people with autism scored much higher on the scale than people with psychiatric disorders. Additionally, respondents were asked about any diagnosis given by a psychiatrist and were not screened with any additional questionnaires or interviews. Diagnoses more objective than self-description should be implemented to increase the value of results.

A different pattern was found in the group of musicians—they scored significantly lower on the scale than non-musicians. Previous findings on this subject have been inconsistent. Some results have suggested a more frequent prevalence of sound sensitivities (not misophonia) among musicians [24,25] while others did not reveal any difference between musicians and non-musicians in terms of levels of sensitivity to sounds [23]. Presumably, musicians can perceive sounds differently to non-musicians, as well as questions on sound issues. Nonetheless, the sample, especially of musicians and people with autism, cannot be representative because of the methodology (internet-based, non-randomized study), so the results must be interpreted with caution.

Finally, the study also provides valuable information on types of emotions related to misophonia. In the second version (with 21 items; not the final version) of the questionnaire, there were two items concerning the characteristics of emotions: a general one (“When I hear some …. immediately I experience *unpleasant emotions”*) and a specific one (“...immediately I experience *anger*”). The question specifying anger was characterized by better values in the EFA analysis than the question which mentioned general emotions. Moreover, the IRT analysis showed that fewer people answered this question according to the key (i.e., agreed with the item statement), so it was more specific for people with misophonia. This is another proof that misophonia is related to the strong experience of anger, not to general unpleasant emotions.

### Limitations

Despite the good psychometric values of MisoQuest, this study has certain limitations that should be taken into consideration when using the tool, as well as in the further development of questionnaires for measuring misophonia.

Firstly, the criterion validity was based on a self-report assessment of the symptoms of misophonia. This was due to the inability to individually investigate a large number of people with specific, yet not officially recognized, symptoms. We also rejected the idea of calling participants on the phone because it would not be possible to properly take care of the participants’ wellbeing and safety. The questions used for assessing criterion-related validity were derived from the diagnostic criteria on which MisoQuest was based. Nonetheless, the items in MisoQuest are slightly broader than the direct questions taken from the diagnostic criteria.

We did not test the convergent validity of MisoQuest by correlating its results with other misophonia measures because of some methodological issues which we believe are critical for proper measurement of this construct.We decided not to use A-MISO-S because its main purpose is to measure the intensity of clinical symptoms of misophonia when it is already known to be present. The MQ was also unsuitable for testing the convergent validity of our measure because it detects a much wider construct than does MisoQuest. In MisoQuest, the emotional reaction must be immediate (in MQ there are items on repetitive sounds, in MisoQuest anger must be present), in MQ sadness or annoyance are sufficient to be accepted as misophonic reactions. Thus, it would not be informative to compare MisoQuest to MQ in order to assess its convergent validity because these two questionnaires are based on different diagnostic criteria. Moreover, at the time of performing this study, there was no Polish version of MQ available.

No hyperacusis questionnaire was used in order to assess MisoQuest’s divergent validity because there is still no consensus on the comorbidity of these two disorders. It is highly possible that a large number of people experience both, so the correlation between results from those questionnaires would not be informative either. If the results showed that there was a high correlation between MisoQuest and a questionnaire that measures hyperacusis, then it would not be possible to conclude whether MisoQuest diagnosed hyperacusis rather than misophonia or whether people with misophonia also suffer from hyperacusis. More meticulous discriminant validity, with more objective methods such as audiological assessment of Loudness Discomfort Level and interviews focused on hyperacusis, should be used in order to make sure that the questionnaire excludes people who have hyperacusis but not misophonia. Additionally, it was not possible to perform discriminant analysis with hyperacusis questionnaires because there has been no Polish version of any hyperacusis questionnaire available until now.

Further analyses for external validity of MisoQuest based on psychophysiological, psychiatric, and audiological indicators are in progress.

Secondly, aversive triggers were limited to sounds; while some data show that misophonia might be not limited to sound triggers, visual triggers are also commonly reported to be aversive for some people with misophonia [5,8,38]. This might also be seen as an advantage of the questionnaire, as the significance of the role of selective sensitivity in other modalities has not been as explored as sound sensitivity and could influence the results. Nonetheless, the role of other modalities in misophonia such as sight, touch, or smell should also be studied in the future.

Thirdly, there was a limited number of people with a second diagnosis, which made it impossible to perform statistical analysis for each of the diagnoses. The grouping of people with affective and anxiety disorder with people with autism into one variable is a severe drawback of the study. The results indicate a need to explore characteristics of specific sound sensitivities in groups with particular disorders in further studies.

Fourthly, there is no information provided by the study on the exact sounds that were considered to be triggers. However, this was not the aim of the study because of a vast variety of possible triggers in misophonia; this should be explored in further studies.

Fifthly, MisoQuest was developed in Polish and tried out on a Polish population. Therefore there is a need for further validation and adaptation for other countries and different languages.

Lastly, this questionnaire was constructed and validated in an internet-based, self-report study, on people recruited via social media. Thus, even though an effort was made to advertise the study in a variety of internet groups and forums, including the use of paid Facebook advertisements, and to recruit participants from various sources, there is a need for further validation of MisoQuest. Additional validation of MisoQuest with psychological, audiological and experimental assessments is ongoing.

## 5. Conclusions

We have created MisoQuest, a questionnaire which assesses misophonia. It was developed and tested in Poland and the Polish version is characterized by good psychometric values. MisoQuest is substantially different from the misophonia questionnaires currently in use. It is also the first fully validated misophonia questionnaire. MisoQuest results suggest that certain triggers might be more specific to misophonia and that people with misophonia do not respond to misophonic triggers in a violent way.

## Figures and Tables

**Figure 1 ijerph-17-01797-f001:**
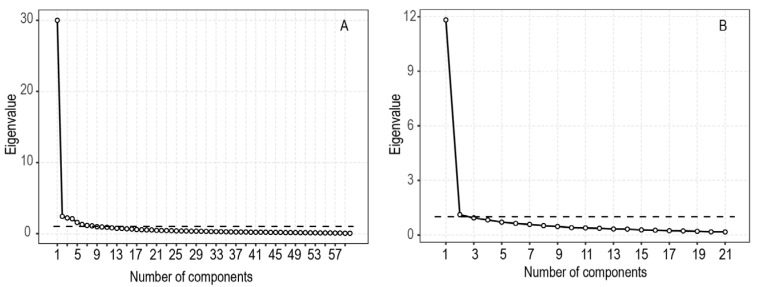
The scree plots from both phases of the development of MisoQuest: (**A**) the first phase, in which 60 items were used; (**B**) the second phase, in which 21 items were used.

**Figure 2 ijerph-17-01797-f002:**
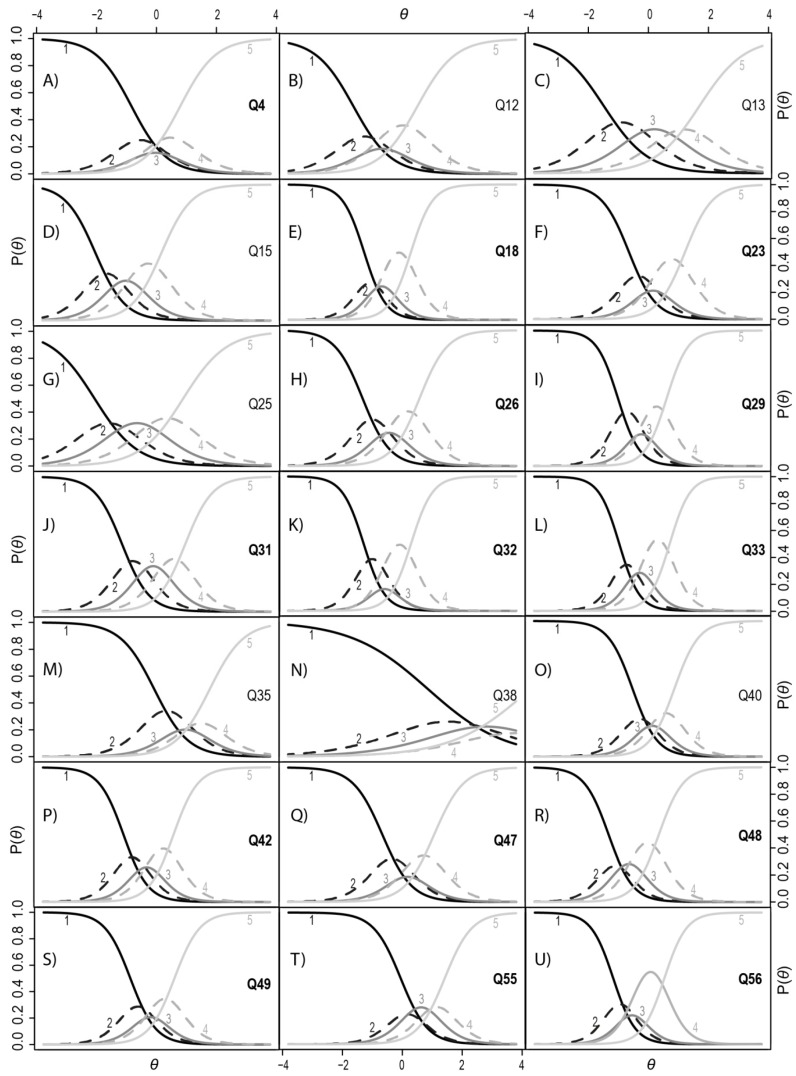
Trace curves for the items of MisoQuest. Items added to the final questionnaire are in bold (i.e., Q4). The functions describe, in probabilistic terms, how likely a person with a higher standing on a trait (*Ɵ*) is to provide a response in a different category to a person with a low standing on the trait. There are 21 plots, one for each item. Five curves on the plot represent each possible response to the item. Curve 1 represents the response “strongly disagree”, curve 2 the response “disagree”, curve 3 the response “neither agree nor disagree”, curve 4 the response “agree” and curve 5 the response “strongly agree”. The person’s misophonia level is denoted by *Ɵ*, and is plotted along the horizontal axis. The vertical axis shows the probability P(*Ɵ*) of each response given the person’s level of misophonia. The curve for option 1 is high at the lowest ability levels and gradually declines with higher levels of misophonia. Similarly, the probability of response 5 is very small at low misophonia levels (*Ɵ*) but rises as *Ɵ* increases. The probability of the other options rises with *Ɵ* to a certain point and then declines again.

**Figure 3 ijerph-17-01797-f003:**
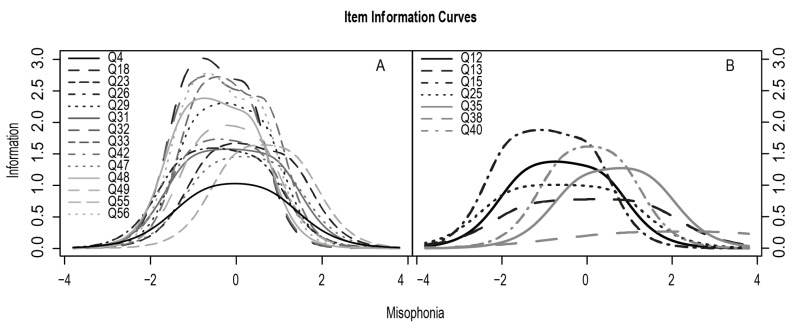
(**A**) Item information curves for MisoQuest and for (**B**) items that were rejected.

**Table 1 ijerph-17-01797-t001:** Factor Loadings (CFA), Test-Retest Reliability, and Discrimination Level from the IRT Model for the MisoQuest are shown. The Items on the Final Version of MisoQuest have been shaded gray.

		Phase 1	Phase 2			
Domain of the Diagnostic Criteria	MisoQuest items (numbers of items from the I version_II version)	Factor loadings	Factor loadings	Item-scale correlations	Test-retest reliability (ICC)	IRT unconstrained model Discrimination
Reaction to Specific Sound	Q4_16 “I find some sounds made by the human body unbearable.”	0.698	0.718	0.69	0.77	1.685
	Q12_1 “I find some quiet sounds unbearable.”	0.703	0.701	0.67	0.57	1.592
	Q13_6 “I find sounds made by things/items unbearable.”	0.355	0.652	0.62	0.54	1.343
Occurrence of Reaction	Q15_2 “When I hear some sounds, I am immediately overcome by unpleasant emotions.”	0.827	0.725	0.69	0.57	2.010
	Q18_19 “Some unpleasant sounds make me instantly angry.”	0.834	0.805	0.78	0.67	2.763
Emotional Reactions	Q23_12 “I start feeling anger the moment I see a thing/animal/person that might make an unpleasant sound at any time.”	0.825	0.778	0.75	0.70	2.202
	Q25_3 “When I hear an unpleasant sound, my heart starts beating faster.”	0.757	0.646	0.61	0.69	1.397
	Q26_9 “When I hear unpleasant sounds, I start sensing emotions in my body (e.g., I sweat, feel pain, feel pressure, my muscles tense).”	0.646	0.769	0.74	0.73	2.165
	Q29_5 “Unpleasant sounds make me feel overwhelmed.”	0.818	0.841	0.82	0.68	2.794
	Q31_7 “I become anxious at the mere thought of an unpleasant sound.”	0.862	0.773	0.74	0.67	2.201
Control of Emotional Reactions	Q32_8 “Some sounds bother me so much that I have difficulty controlling my emotions.”	0.857	0.818	0.79	0.59	2.806
	Q33_14 “I put a lot of effort into controlling emotions when I hear an unpleasant sound.”	0.834	0.856	0.83	0.66	2.943
	Q35_10 “When I hear unpleasant sounds, my eyes start filling with tears.”	0.832	0.664	0.64	0.71	1.764
	Q38_11 “As a child I would hit people who made unpleasant sounds.”	0.851	0.402	0.37	0.62	0.779
Attitude to Own Reactions	Q40_13 “I think that something is wrong with me because I react too strongly to certain sounds.”	0.844	0.797	0.77	0.71	2.435
	Q42_4 “I believe that my reactions to sounds are exaggerated but I can’t get rid of them.”	0.828	0.807	0.79	0.77	2.450
Avoidance	Q47_15 “If I can, I avoid meeting with certain people because of the sounds they make.”	0.812	0.760	0.73	0.69	2.041
	Q48_17 “I feel that my mental state worsens if I cannot leave a place where there’s an unpleasant sound.”	0.786	0.789	0.76	0.71	2.460
	Q49_18 “I often think about how to drown out unpleasant sounds.”	0.736	0.781	0.76	0.72	2.344
Daily Functioning	Q55_20 “I am scared that unpleasant sounds may impact my future.”	0.846	0.739	0.71	0.78	2.210
	Q56_21 “When meeting with other people, I am sometimes irritated because of unpleasant sounds that are present.”	0.804	0.805	0.78	0.69	2.699

**Table 2 ijerph-17-01797-t002:** Demographic characteristics of the studied samples.

Variable	Phase 1 (*N* = 383)	Phase 2 (*N* = 322)
Age, Mean ± Standard Deviation (years)	31.7 ± 9.22	31.7 ± 8.83
Gender,		
Male, No. (%)	53 (13.8%)	65 (20.2%)
Female, No. (%)	330 (86.2%)	257 (79.8%)
Educational Attainment		
Lower Secondary Education	-	5 (1.6%)
High School Education	-	81 (25.2%)
Non-tertiary Education	-	1 (0.3%)
Higher Education	-	232 (72%)
Place of Residence		
Village	-	28 (8.7%)
Small City (less than 500k residents)	-	88 (27.3%)
Large City (more than 500k residents)	-	206 (64%)
Marital Status		
Single, Never Married	-	88 (27.3%)
Cohabiting with Partner	-	110 (34.2%)
Married	-	105 (32.6%)
Divorced	-	9 (2.8%)
Diagnosis, No. (%)		
Misophonia	68 (17.8%)	63 (19.6%)
Hyperacusis	-	11 (3.4%)
Autism/Asperger Syndrome	-	33 (10.2%)
Affective Disorders	-	16 (5%)
Anxiety Disorders	-	4 (1.2%)
Personality Disorders	-	12 (3.7%)
Neurological Disorders	-	5 (1.5%)
Musician, No. (%)	-	96 (29.8%)

**Table 3 ijerph-17-01797-t003:** Criterion-related validity of MisoQuest.

Group (*n*)	MisoQuest M(SD)	t Test (df)	*p*-Value Cohen’s d
With Misophonia (*n* = 61)	65.72 (4.3)	t = 21.65 (308.85)	*p* < 0.001 d = 2.13
Without Misophonia (*n* = 254)	41.77 (15.3)		
Without Diagnosis (*n* = 253)	44.83 (17.1)	T = 3.9 (111.322)	*p* < 0.001 d = 0.52
With Diagnosis (*n* = 62)	52.87 (13.84)		
-ASD (*n* = 31)	53.39 (12.49)		
-Hyperacusis (Without Misophonia) (*n* = 8)	52.87 (15.35)		
-Psychiatric Diagnosis (*n* = 23)	52.17 (15.58)		
Musician (*n* = 95)	43.55 (13.9)	t = 2.51 (225.67)	*p* < 0.05 d = 0.29
Non-Musician (*n* = 220)	47.64 (17.78)		

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
