# Peer review of "Development and Psychometric Properties of MisoQuest—A New Self-Report Questionnaire for Misophonia"

_ijerph, 2020, doi:10.3390/ijerph17051797_

Round 1

Reviewer 1 Report

Thank you to the editor and the authors for the opportunity to review this manuscript. The authors validated a modified misophonia questionnaire – MisoQuest - to quantify the severity of misophonia.

In the introduction strong arguments were made for developing a new questionnaire with a detailed analysis of the current questionnaires on misophonia published in the literature. In my opinion, the methodology and analysis of the result is statistically sound, although I am not an expert in statistics. The discussion is well-formulated with an in-depth analysis and elaboration of rationale behind the construct of the individual elements of the questionnaire.

The language and style of the manuscript can be polished to be more appropriate to the medium – the authors can consider enlisting the help of a native English speaker colleague. Overall, this scientifically-sound manuscript will serve as an important research contribution to the nascent field of misophonia.

Minor feedback

  1. Line 41 – ‘some researchers noted that the sound… or loud voices can also trigger misophonic reactions,’ I think Quek et al. noted high-pitched voice instead of loud (volume) voices to evoke misophonic reactions in patients.

  1. Line 47-48 – ‘In misophonia, according to the current state of knowledge, the acoustic  characteristics of sounds have no impact on emotional arousal.’ – I think it would be good to provide a reference to support this claim.
  2. Line 48-49 – ‘The type of aversive triggers are different among each individual’

  1. Line 51 – ‘Misophonia is not yet…’

  1. Line 348 – ‘… and anger as a core’

  1. Line 376-383 – This section of the manuscript discussed the lack of association between ‘childhood misophonia’ and acts of violence during childhood. This was done by asking subjects if they had commited acts of violence when they were exposed to trigger sounds when they were young. However, I think we cannot conclusively conclude that there was even the presence of ‘childhood misophonia’ in the past since no attempt was made on quantifying the severity of misophonia symptoms when subjects were younger. Secondly, the authors can specify which age the subjects are made to recall their acts of violence from – ‘childhood’ is a very vague time.

Reviewer 2 Report

Nice article; please see comments and suggestions in attached article

Reviewer 3 Report

Thank you for inviting me to review the paper on “Development and Psychometric Properties of MisoQuest - New Self-report Questionnaire for Misophonia”. I welcome more paper in the field of misophonia. I have the following recommendations to improve this paper and happy to review this paper again.

  1. Line 35, the authors stated, “The dominant, but not exclusive, emotions related to misophonia are anger, disgust, extreme irritation, and anxiety [4,5,6,].” I read the references. Some of the references are not related to anxiety. The authors should include the following paper published by IJERPH.

Quek TC, Ho CS, Choo CC et al. Misophonia in Singaporean Psychiatric Patients: A Cross-Sectional Study. Int J Environ Res Public Health. 2018;15(7):1410.

  1. It is rather unusual to present a table under the method. Please move Table 1 to the result section.

  1. Under the statistical analysis, line 175, the authors stated: “Confirmatory Factor Analysis (CFA) was abandoned.” This statement appeared suddenly and our readers may not be familiar with CFA. Please add an additional statement to explain what CFA is.

Line 174: Confirmatory factor analysis (CFA) is usually performed to validate the potential models of a new questionnaire obtained by EFA in the literature. Because of the final form of MisoQuesthad only one factor and very high ….

Reference:

Mak KK, Lai CM, Ko CH, et al. Psychometric properties of the Revised Chen Internet Addiction Scale (CIAS-R) in Chinese adolescents. J Abnorm Child Psychol. 2014;42(7):1237–1245. doi:10.1007/s10802-014-9851-3

  1. The authors need to check Table 2. The fonts are too big and part of the table seems to be cut.

Round 2

Reviewer 3 Report

Thanks for the amendments. I recommend publication.